DNA metabarcoding of zooplankton communities: species diversity and seasonal variation revealed by 18S rRNA and COI

Zhao Lina
Zhang Xue
Xu Mengyue
Mao Ying
Huang Yuan yuanh@snnu.edu.cn
College of Life Sciences, Shaanxi Normal University , Xian , Shaanxi , China
Wangensteen Owen
Electronic publication date: 2021 Mar 19
Publication date: 2021
Volume: 9
Electronic Location ID: e11057
Received 2020 Jun 25; Accepted 2021 Feb 12
Copyright: ©2021 Zhao et al.
Copyright year: 2021
Copyright holder: Zhao et al.
License: This is an open access article distributed under the terms of the Creative Commons Attribution License, which permits unrestricted use, distribution, reproduction and adaptation in any medium and for any purpose provided that it is properly attributed. For attribution, the original author(s), title, publication source (PeerJ) and either DOI or URL of the article must be cited.
License URL: https://creativecommons.org/licenses/by/4.0/

Keywords: Zooplankton diversity, Metabarcoding, Seasonal variation, Water environmental factor, Sanmenxia reservoir

Funding: The National Natural Science Foundation of China No. 31872217 This work was supported by the National Natural Science Foundation of China (No. 31872217). The funders had no role in study design, data collection and analysis, decision to publish, or preparation of the manuscript.

==============================
Background

Zooplankton is an important component of aquatic organisms and has important biological and economical significance in freshwater ecosystems. However, traditional methods that rely on morphology to classify zooplankton require expert taxonomic skills. Moreover, traditional classification methods are time-consuming and labor-intensive, which is not practical for the design of conservation measures and ecological management tools based on zooplankton diversity assessment.

Methods

We used DNA metabarcoding technology with two different markers: the nuclear small subunit ribosomal RNA (18S rRNA) and mitochondrial cytochrome c oxidase (COI), to analyze 72 zooplankton samples collected in 4 seasons and 9 locations from the Sanmenxia Reservoir. We investigated seasonal changes in the zooplankton community and their relationship with water environmental factors.

Results

A total of 190 species of zooplankton were found, belonging to 12 phyla, 24 classes, 61 orders, 111 families, and 174 genera. Protozoa, especially ciliates, were the most diverse taxa. Richness and relative abundance of zooplankton showed significant seasonal changes. Both alpha and beta diversity showed seasonal trends: the diversity in summer and autumn was higher than that in winter and spring. The zooplankton diversity was most similar in winter and spring. By correlating metabarcoding data and water environmental factors, we proved that water temperature, chemical oxygen demand, total nitrogen and ammoniacal nitrogen were the main environmental factors driving the seasonal changes in zooplankton in the Sanmenxia Reservoir. Water temperature, followed by total nitrogen, were the most influential factors. This study highlights the advantages and some limitations of zooplankton molecular biodiversity assessment using two molecular markers.

Introduction

Protecting the biodiversity of freshwater ecosystems is particularly important in the 21st century. Freshwater ecosystems not only contain the most concentrated biodiversity of all ecosystems on the planet but are also the most endangered ecosystems worldwide (Dudgeon et al., 2006). These ecosystems are susceptible to anthropogenic factors, such as climate change, eutrophication, and the introduction of nonnative species, all of which will lead to a reduction in biodiversity (Dudgeon et al., 2006; Revenga et al., 2005; Vörösmarty et al., 2010). Over the past few decades, the biodiversity of freshwater ecosystems has decreased significantly more than that of terrestrial or marine ecosystems (Sala et al., 2000). Although data on the extinction rate of freshwater ecosystems are limited to North America (Ricciardi & Rasmussen, 1999), these data hint at a global crisis of freshwater biodiversity (Abell et al., 2008). Therefore, protecting the biodiversity of freshwater ecosystems is a key priority. However, protection requires a detailed understanding of this biodiversity.

Zooplankton, including protozoans, rotifers and microcrustaceans (cladocerans and copepods) (Allan, 1976; Pace & Orcutt Jr, 1981), have important biological and economic value in freshwater ecosystems. Not only are zooplankton the major consumers in aquatic ecosystems, they are also an important food source for fish larvae and, thus, have an important role in connecting primary producers with higher nutritional consumers (Allan, 1976; Banse, 1995; Lubzens, 1987). Moreover, due to the influence of abiotic factors (such as water temperature) and seasonal changes in biological factors, zooplankton communities also exhibit seasonal change patterns. In addition, zooplankton are very sensitive to external disturbances, and they are commonly used as indicators of water quality (Branco et al., 2002; Gannon & Stemberger, 1978; Oh et al., 2017; Sládeček, 1983). For example, Brachionus calyciflorus can be used as an indicator of eutrophic water bodies (Gannon & Stemberger, 1978), which has far-reaching importance in water environment monitoring. However, the fragility of the zooplankton itself, the small size, the difficulty of obtaining observations, and the species variety of zooplankton have limited the understanding of zooplankton diversity (Machida et al., 2009), which is not conducive to the protection of aquatic biodiversity. Among freshwater ecosystems, reservoirs have always been a subject of interest. Reservoirs enable river systems to be strongly regulated while providing the convenience of hydropower for human production activities (Yang & Lu, 2014). However, damming considerably perturbs the river’s export of organic carbon to the ocean (Gao et al., 2019; Maavara et al., 2017). In addition, the water residence time of the storage reservoir is significantly increased and the flow velocity is obviously decreased after dam closure (Friedl & Wüest, 2002), which hinders the movement of aquatic organisms, and alters their habitats thus has a significant effect on aquatic biodiversity (Vörösmarty et al., 2010). Direct and indirect habitat changes are a major factor driving the decline or extinction of freshwater biological species, especially for species with limited diffusion capacity (Pimm et al., 2014; Revenga et al., 2005), such as zooplankton.

Although zooplankton have important biological and economic value in freshwater ecosystems, traditional taxonomic methods used to characterize zooplankton communities are not sufficient for large-scale biodiversity research. The limitations of traditional morphological identification methods cannot be completely overcome in the short term (Trebitz et al., 2017). On the one hand, traditional morphological classification methods are not only time consuming and labor intensive but also lack resources for species identification (such as increasingly limited funding to support taxonomic scientists for scientific research) (Creer et al., 2016; Pawlowski et al., 2016). On the other hand, traditional morphological identification methods require scientific researchers to be proficient in taxonomy, but academia currently lacks experienced taxonomists (Pawlowski et al., 2016). In addition, a large amount of classification research work may also contain classification errors (Sundermann et al., 2005). To overcome the limitations of conventional morphology-based methods, improved methods are urgently needed to detect zooplankton species in freshwater ecosystems more quickly, sensitively, and effectively (Xiong et al., 2020). DNA metabarcoding technology can quickly alleviate this situation and has proven to be a powerful tool for large-scale biodiversity research (Beng et al., 2016; Ji et al., 2013; Shokralla et al., 2012). This technology assigns ecological value to the taxonomic operational units (OTUs) generated by high-throughput sequencing data and quickly, economically and efficiently realizes the classification and identification of environmental DNA (Fernando, 2002; Pawlowski et al., 2016), making an important and unique contribution in the field of ecology (Xie et al., 2017; Zhang et al., 2018). DNA metabarcoding technology provides detailed information about zooplankton biodiversity and can support biodiversity conservation (Xiong et al., 2020). In addition, it has been successfully used to track seasonal changes in the zooplankton community (Berry et al., 2019; Mwagona, Chengxue & Hongxian, 2018; Nandini, Merino-Ibarra & Sarma, 2008; Tan et al., 2004; Vanderploeg et al., 2012). Metabarcoding technology sensitively reflects the relationship between the abundance of the zooplankton community and environmental factors and provides methodological support for exploring the water environmental factors that drive seasonal changes in zooplankton (Yang et al., 2017a). Most biodiversity studies based on DNA metabarcoding technology use a single molecular marker method (Amaral-Zettler et al., 2009; Bucklin et al., 2019; Chain et al., 2016), but the taxon scope of the study of single molecular markers is limited (Drummond et al., 2015; Giebner et al., 2020). However, the multigene-based metabarcoding method can increase the taxonomic breadth, which is especially important for assessing the biodiversity of highly diverse groups such as zooplankton.

Considering the dramatic changes in ecology, water transparency, nutritional status, the densities in the downstream Sanmenxia Reservoir and the strong dependence of zooplankton richness on the lake conditions of the dam (creating a stable environment for the survival and reproduction of different zooplankton populations), we hypothesize that the Sanmenxia Reservoir has a particularly rich zooplankton community. Moreover, due to seasonal changes in the water environment, the zooplankton community is likely to exhibit a seasonal change pattern in the reservoir, and this seasonality is closely related to water environmental factors. However, it is not clear which water environmental factors have an impact on the richness and relative abundance of the zooplankton community in the Sanmenxia Reservoir or which factors have the strongest influence. To verify the above hypothesis, we attempt to characterize the zooplankton diversity and water environmental factors in the Sanmenxia Reservoir by obtaining seasonal (summer, autumn, winter and spring in July 2017, October 2017, January 2018 and March 2018, respectively) zooplankton samples and water samples. We used a multigene approach with two independently evolved markers,the nuclear small subunit ribosomal RNA (18S rRNA v4 region; Hadziavdic et al., 2014) and and the mitochondrial cytochrome c oxidase (COI; Leray & Knowlton, 2015). The main purposes of our study are (1) to explore the composition of the zooplankton community in the Sanmenxia Reservoir, (2) to reveal the dynamics of seasonal changes in the zooplankton community, and (3) to determine the main water environmental factors that drive the seasonal changes in zooplankton diversity in the reservoir area.

Materials and Methods

Study area, zooplankton sample collection and measurement of water environmental factors

The Sanmenxia Reservoir, which is located in the lower reaches of the Middle Yellow River basin in China, was effectively completed in 1960. The reservoir is the first large-scale water conservancy project built on the Yellow River and controls 89% of the water in the Yellow River basin (Chen et al., 2017; Wang, Wu & Wang, 2005). The reservoir area is in a warm temperate continental monsoon climate. The interannual temperature fluctuates within the range of −18.3 °C∼42.3 °C, and the average annual temperature is 14.1 °C (Zhang et al., 2012). According to the geographical features of the Sanmenxia Reservoir and the field investigation, nine sampling points (as 9 biological replicates in this study) with representative habitats (the front and rear of the dam of Sanmenxia Reservoir) were set up in the reservoir area to collect samples (three replicates each site) in summer, autumn, winter and spring in July 2017, October, 2017, January 2018 and April 2018, respectively (Fig. S1). Geographical coordinates of the sampling site were collected with a global positioning system (GPS) (Garmin Legend, Garmin USA). Zooplankton samples were collected by filtering 20 L of water using plankton nets (46-µm mesh). The plankton net was dragged along the water surface repeatedly in a “∞” shape at a depth of 0.3 m for 5 min to collect and concentrate samples. Then, the samples were transferred into a 50 mL centrifuge tube and immediately preserved in 95% ethanol. In the laboratory, the zooplankton samples were further filtered through 0.22 µm microporous filter paper (Whatman, UK) to remove excess water, placed in a five mL centrifuge tube, and then stored at −80 °C until DNA extraction could be completed.

During the zooplankton sampling period, additional 500 mL water samples were collected at each sampling site to measure chemical oxygen demand (COD), total nitrogen (TN), total phosphorus (TP), ammonia nitrogen (NH4-N) and pH. These measurements were taken in the laboratory of Shaanxi Normal University. Water temperature (WT) and dissolved oxygen (DO) were measured directly in the field using a BANTE820 Portable Dissolved Oxygen Meter (Bante, China).

DNA extraction, PCR amplification, library construction and high-throughput sequencing (HTS)

The total DNA of the zooplankton community was purified by mixing DNA isolated from three samples from each sampling site using the DNeasy Blood and Tissue Kit (Qiagen, Germany). The three biological replicates of each sampling site were pooled to yield more DNA. DNA quality was monitored using a NanoDrop 2000 (Thermo Scientific, USA) and 1% agarose gel electrophoresis. Mitochondrial cytochrome c oxidase I (COI) was amplified by using the universal COI primers mlCOIintF and jgHCO2198 (Geller et al., 2013; Leray & Knowlton, 2015), and the V4 region of the nuclear small subunit ribosomal RNA (18S rRNA) was amplified by using HPLC-purified PCR primers (Nolte et al., 2010) (Table S1). PCR was performed with 25 µL reactions for each sample, including 1 µL of DNA template (30 ng/µL), 1 µL of primers (10 µM), 19.9 µL of ultrapure water, 2.5 µL of High-Fidelity PCR buffer (10×), 0.5 µL of dNTP mix (10mM) and 0.1 µL of Platinum Taq DNA polymerase (Invitrogen, USA). The PCR reaction procedure was as follows. 18S: 95 °C for 3 min; 35 cycles at 95 °C for 45 s; 50 °C for 50 s; 72 °C for 45 s; and 72 °C for 10 min; COI: 95 °C for 10 min; 16 cycles at 94 °C for 10 s; 62 °C for 30s (decreasing by 1 °C per cycle); 68 °C for 60 s; 25 cycles at 94 °C for 10 s; 46 °C for 30 s; 68 °C for 60 s; 72 °C for 10 min extension; and 4 °C insulation. Then, the PCR products of the same sample (three PCR replicates per sample) were mixed and subjected to 2% agarose gel electrophoresis. PCR product was recycled by cutting the gel using an AxyPrepDNA gel extraction kit (Axygen Biosciences, USA), and subjected to Tris_HCl elution and 2% agarose electrophoresis. After purification, a standardized procedure for qualified PCR products was followed to form an amplicon library, and a MiSeq Reagent Kit v2 (500 cycle) was used to perform high-throughput sequencing of 2 ×250 bp reads on the Illumina MiSeq platform (Allwegene, China).

Data processing

Initial quality control was completed using Trimmomatic (Bolger, Lohse & Usadel, 2014), a 50 bp window size was set, and bases with average Phred quality score less than 20 were trimmed. The primer sequence was removed using Cutadapt (Martin, 2011). Merging of raw paired-end reads was conducted using FLASH (Magoč & Salzberg, 2011) with more than 10 bp in the minimum overlap sequences, and the maximum allowable mismatch ratio not exceeding 0.1. Recognition and removal of chimeric sequences were conducted using UCHIME (Edgar et al., 2011). OTU clustering was performed using UPARSE (namely, USEARCH was used to cluster OTUs according to 97% similarity sequences, excluding single sequences, obtain representative sequences and then map all their sequences according to 97% similarity to OTUs to form an OTU list) pipeline (Edgar, 2013). OTU annotation was conducted on the representative sequences of each OTU against Silva (18S rRNA) and the BOLD reference database (COI) (Altschul et al., 1990; Pruesse et al., 2007; Ratnasingham & Hebert, 2007). The species names of OTUs were designated as sequence identity exceeding 97%, and the genus names of OTUs were designated as sequence identity between 90% and 97%. If an OTU could not be assigned to a species, it was assigned to a higher taxonomic level. We also removed all bacteria, fungi, and Viridiplantae from the OTU annotation tables to focus exclusively on zooplankton for the purpose of this study.

Zooplankton diversity analysis

Statistical analyses were performed by using the “vegan” package in R v.3.6.3 (Oksanen et al., 2010; R Core Team, 2013). Rarefaction curves were used to measure the saturation of zooplankton samples. Venn diagrams were used to show the number of different taxa recovered by using 18S (blue) or COI (yellow) at different taxonomic levels. Bar charts were used to reflect the difference in OTU richness and relative abundance produced by DNA metabarcoding technology in 18S (blue) or COI (yellow). Bubble plots were used to reflect the composition and seasonal variation in the zooplankton community. Alpha diversity was characterized by the Shannon and ACE indices, which were drawn by box plotting. In this process, since the data did not obey a normal distribution, the Kruskal–Wallis nonparametric test was used to determine whether seasonal changes in zooplankton diversity were significantly different. Subsequently, since the OTU richness data obeyed a normal distribution and homogeneity of variance, analyses of variance (ANOVA) were selected and performed using Tukey’s honest significant difference (Tukey HSD) tests to analyze whether the richness of each zooplankton group differed significantly among seasons. Bar charts were also used to characterize the seasonal changes in the relative abundance of each zooplankton group, and the Welch t test was used to determine whether their seasonal changes were significant in adjacent sampling seasons. To evaluate the seasonal variation in the zooplankton community between samples, a principal coordinate analysis (PCoA) was used to cluster the samples by sampling season based on the Jaccard dissimilarity index and Bray–Curtis dissimilarity index, and pairwise comparisons of all sampling seasons were performed using PERMANOVA (Anderson, 2001). An analysis of similarity (ANOSIM) was also used for further evaluation. According to the characteristics of the measured water environmental factor data, the relationship between the variables was analyzed using Spearman’s correlation analysis with Bonferroni correction to account for multiple testing. To investigate the water environmental factors that drive seasonal changes in the zooplankton community, the OTU data matrix was related to the water environmental factor matrix by using the partial Mantel test, and relative abundance data were combined with water environmental factors for canonical correspondence analysis (CCA). An Adonis analysis (permutations: 999) was used to test the contribution (variation, R2) and significance of each water environmental factor.

Results

Results of amplicon sequencing

We metabarcoded a total of 36 samples for each marker (nine locations per season, for a total of four seasons). After the Illumina sequence passed all filtering procedures, 2,185,317 and 910,137 high-quality reads (including non-plankton metazoans) were obtained from the 18S rRNA and COI metabarcoding, respectively. In the 18S rRNA dataset, the sequencing depth per sample ranged from 5510 to 116,411, with an average of 60,703. In the COI dataset, the sequencing depth per sample ranged from 7057 to 68,313, with an average of 25,282 (Table S2). The OTUs of single reads were excluded from further analysis to reduce the possible effects of spurious signals. This exclusion could conservatively remove potential misamplified fragments, although they may represent very low-abundance zooplankton groups. The rarefaction curves of 72 zooplankton samples stabilized with increasing sequencing depth, which indicated that most of the sampling points were properly sampled and that the data tended to be saturated (Fig. S2). DNA metabarcoding of zooplankton samples resulted in the identification of 495 OTUs from the 18S rRNA gene, and 1099 OTUs from the COI gene. Approximately 5.25% (26 of 495 OTUs) observed in the 18S rRNA datasets could only be identified to kingdom and were labeled “Eukaryota” or “Metazoa”. Approximately 3.46% and 0.54% (38 and 6 of 1099 OTUs, respectively) were labeled “Invertebrate environmental sample” and “Arthropoda environmental sample” in the COI datasets, respectively. Approximately 64.44% (319 of 495 OTUs, 18S rRNA) and 35.94% (395 of 1099 OTUs, COI) of the OTUs were classified as zooplankton, and the remaining OTUs were annotated as Insecta, Gastropoda, etc. (Table S3). The number of zooplankton OTUs ranged from 26 to 148 (18S rRNA) and 49 to 146 (COI) in each sample (Table 1).

Zooplankton community composition

There were differences in the number of zooplankton taxa identified by the two markers at different taxonomic levels (Table 2, Fig. 1). A total of 12 phyla were identified in this study, including 24 classes, 61 orders, 111 families, 174 genera and 190 species. Of the 319 nonsingleton zooplankton OTUs in the 18S rRNA gene, 12 phyla were represented across 23 classes, 54 orders, 83 families and 110 genera (Table 2). Of the 395 nonsingleton zooplankton OTUs in the COI gene, 4 phyla were represented across 7 classes, 19 orders, 42 families and 80 genera (Table 2). At the phylum level, the four phyla detected by the COI were also be recovered with 18S. The corresponding percentages were 85.71% for class, 63.16% for order, 33.33% for family, 20.00% for genus, and 9.4% for species-level taxa (Fig. 1).

Table 1 Alpha diversity of the entire zooplankton assemblages in each sample.

Samples	18S		COI	
	OTUs	reads	Shannon	ACE		OTUs	reads	Shannon	ACE	
SUM_1	101	31787	1.73	110.83		130	22340	2.6	155.41	
SUM_2	142	58326	2.1	160.11		139	22131	2.76	156.38	
SUM_3	144	56263	1.95	170.04		135	21489	3.08	141.1	
SUM_4	131	42619	2.23	143.83		145	20755	2.85	169.79	
SUM_5	145	87908	2.16	174.61		119	21384	2.58	137.52	
SUM_6	148	84124	2.27	179.04		122	21456	2.62	147.47	
SUM_7	81	48914	1.45	92.75		136	19656	2.58	164.43	
SUM_8	99	34553	1.59	118.58		98	18675	2.34	115.29	
SUM_9	104	51766	1.57	123.03		96	18814	2.3	110.99	
AUT_1	123	30108	1.67	168.92		112	5882	2.81	140.85	
AUT_2	106	18216	1.54	136.19		116	3695	2.87	138.27	
AUT_3	104	27958	1.19	129.64		112	5316	2.83	131.47	
AUT_4	103	34195	1.81	140.86		120	9537	2.85	164.59	
AUT_5	117	44036	1.48	162.35		101	4024	2.72	133.37	
AUT_6	80	26275	2.08	98.48		146	8723	3.09	181.54	
AUT_7	71	52425	1.64	85.9		118	14766	2.85	153.54	
AUT_8	63	69162	1.58	86.26		102	19037	2.57	139.55	
AUT_9	68	66918	1.57	91.2		103	16720	2.55	128.24	
WIN_1	77	31180	1.66	103.39		76	17500	2.03	80.94	
WIN_2	66	16455	1.63	76.45		49	2307	2.27	63.35	
WIN_3	56	11396	2.05	98.19		56	4512	1.72	74.09	
WIN_4	53	71055	1.14	108.61		71	21905	2.31	84.43	
WIN_5	42	55232	1.13	60.79		77	21181	2.06	93.86	
WIN_6	45	28844	1.11	52.05		74	21706	2.02	92.39	
WIN_7	47	8251	1.8	65.28		57	2694	2.64	68.53	
WIN_8	57	8569	1.69	82.21		65	6018	2.52	81.96	
WIN_9	43	12647	1.34	65.47		72	5332	2.7	91	
SPR_1	40	29215	0.74	60.15		66	7367	1.72	78.51	
SPR_2	32	36306	0.64	34.9		59	19461	1.02	67.62	
SPR_3	43	36212	0.87	85.35		79	11795	2.15	97.03	
SPR_4	31	15650	1.36	50.68		87	14348	2.09	114.91	
SPR_5	41	9195	1.6	69.85		78	12331	1.75	86.24	
SPR_6	26	3916	1.84	35.99		79	8985	2.35	82.05	
SPR_7	39	35357	0.92	46.03		68	21177	1.5	99.43	
SPR_8	87	105749	0.92	113.01		85	22711	1.44	114.4	
SPR_9	83	104381	1.12	111.27		93	20955	1.62	131.48	

The richness (Fig. 2A) and species diversity (Figs. 1G; H; Figs. 1I) of the same zooplankton groups showed differences in the results identified by the two markers, with Copepoda richness the only exception (Fig. 2A). Protozoa was the most abundant group in the 18S rRNA, accounting for 68.03% (217 OTUs) and 76.19% (64 species) of zooplankton richness and species diversity, respectively, and only 14.18% (56 OTUs) and 18.80% (22 species), respectively,in COI. Rotifera was the most abundant group (57.47%, 227 OTUs; 51 species, 43.59) in COI but only accounted for 8.46% (27 OTUs) and 11.90% (10 species) of richness and diversity, respectively, in 18S rRNA. Copepoda was the second most abundant taxon between the two markers; its OTUs richness was similar (73, 71OTUs) and 11 and 30 species were identified in the COI and 18S rRNA samples, respectively. The least diverse taxonomic groups was Cladocera (2 OTUs, 18S rRNA; 41 OTUs, COI) using both markers, and 0 and 14 species were identified respectively. In addition, 11 species were identified by both markers (Fig. 1GHI): 2 species of Protozoa, namely, Vermamoeba vermiformis and Reclinomonas americana, 4 species of rotifers, namely, Adineta vaga, Asplanchna sieboldi, Brachionus calyciflorus and Testudinella clypeata, and 5 species of copepods, namely, Megacyclops viridis, Mesocyclops dissimilis, Mesocyclops pehpeiensis, Microcyclops varicans and Pseudodiaptomus inopinus. Brachionus calyciflorus was detected in all (72) zooplankton samples (Fig. S3). Eucyclops macruroides, Sinocalanus sinensis and Brachionidae sp. SHDT150710, and Keratella cochlearis were also present in each zooplankton sample of 18S rRNA or COI genes (Fig. S3).

Table 2 Results of taxonomic assignment of different metabarcoding datasets.

	OTUs	Phylum	Class	Order	Family	Genus	Species	
18S	319	12	23	54	83	110	84	
COI	395	4	7	19	42	80	117	
18S+COI	/	12	24	61	111	174	190	

Figure 1 The number of different taxa recovered by using 18S (blue) or COI (yellow) under different taxonomic levels. (A) Phylum, (B) Class, (C) Order, (D) Family, (E) Genus, (F) species, (G) Protozoa, (H) Rotifera, (I) Copepoda.

Figure 2 The difference in (A) richness and (B) relative abundance recovered in 18S or COI.

The relative abundance of the same zooplankton group differed between the two markers.(Fig. 2B). 18S Copepoda had significantly higher relative abundance than other zooplankton taxa, but 18S Cladocera had the lowest relative abundance. In COI, Rotifera had the highest relative abundance, which was similar to that of Copepoda. However, the relative abundance of Protozoa was low according to both markers.

Seasonal dynamics of zooplankton diversity

The Shannon diversity and ACE index observed in the metabarcoding datasets of each molecular marker were similar, and all showed a maximum in summer or autumn and a minimum in spring or winter (Table 1). Moreover, the entire zooplankton community assemblages showed significant seasonal changes, including the richness estimators ACE, and the diversity estimator Shannon (Fig. 3). The overall trend showed high diversity in summer and autumn and low diversity in winter and spring.

Figure 3 Different diversity estimators show significant differences in seasonal changes in zooplankton community based on different metabarcoding datasets.

(A) 18S Shannon diversity index (B) COI Shannon diversity index, (C) 18S ACE index, (D) COI ACE index.

Figure 4 The richness of zooplankton groups in different seasons shows significant differences based on different metabarcoding datasets.

(A) 18S Protozoa, (B) 18S Rotifera, (C) 18S Copepoda, (D) 18S Cladocera, (E) COI Protozoa, (F) COI Rotifera, (G) COI Copepoda, (H) COI Cladocera. Seasonal changes in zooplankton groups were significant according to ANOVA and multiple comparisons between sampling seasons by using the Tukey HSD test. *, p ≤ 0.05; **, p ≤ 0.01; ***, p ≤ 0.001.

To explore seasonal changes in each zooplankton group, we performed ANOVA on the richness datasets of zooplankton at OTU levels (Fig. 4). The results showed that the richness of zooplankton groups, except for the 18S Cladocera, had significant seasonal changes (Tukey HSD, p < 0.01, Fig. 4), and the species diversity of most zooplankton groups was significantly higher in summer and autumn than in winter and spring. In addition, the trend in seasonal changes in the relative abundance of Rotifera was consistent in the two molecular markers, and Copepoda was no exception. From summer to autumn in 2017, the relative abundance of Rotifera decreased from 21.57% (18S) and 32.78% (COI) to 3.36% and 11.49% , respectively (Figs. 5A; 5D), and then increased to 42.22% and 67.20% in winter 2018 (Figs. 5B; 5E) and finally decreased to 35.20% and 59.92% in spring 2018 (Figs. 5C; 5F). The change tendency of the relative abundance of 18S Protozoa was consistent with that of Rotifera, and the trend was only consistent with that of the COI Protozoa from winter to spring (Fig. 5F). The relative abundance of Cladocera was consistent with the change trend in 18S Copepoda. However, for COI, the relative abundance increased from summer (8.37%) to winter (15.41%) and then decreased in spring, when it reached a minimum (6.31%) (Figs. 5D; 5E; 5F). In fact, the relative abundance of most zooplankton groups, except for the richness, also showed significant seasonal variation between adjacent sampling seasons (Welch t test, p < 0.05). However, the relative abundance of Cladocera showed no significant seasonal changes between adjacent sampling seasons (Welch t test, p > 0.05), although its relative abundance based on COI was obviously higher than that of the 18S rRNA gene.

To assess the seasonal variations in the entire zooplankton community assemblages between samples, we performed a PCoA at the OTU level (Fig. 6). The results showed that within the ellipse with a 75% confidence interval, the samples for most sampling seasons were divided into three distinct clusters. Two distinct clusters were formed separately in summer and autumn, and another large cluster was formed in winter and spring, and there was no overlap between seasons other than between winter and spring (Fig. 6). Thus, the zooplankton communities in winter and spring were more similar to each than were those in the summer and autumn.

In terms of relative abundance, the zooplankton communities during winter and spring were dominated by Rotifera and Copepoda in the 18S metabarcoding datasets, and their relative abundances were 42.22% and 35.20% (Rotifera) and 54.15% and 63.39% (Copepoda) (Fig. 5C), respectively, within which Eucyclops and Brachionus were predominant (Fig. 7A), contributing 36.57% and 33.59% to the similarity of the zooplankton community in winter and spring, respectively. Conversely, the autumn zooplankton samples were nearly all annotated as Copepoda (95.81%) (Fig. 5A), including Eucyclops and Mesocyclops (Fig. 7A). The summer zooplankton samples also had a small number of Rotifera (21.57%), but the abundance was far lower than the relative abundance of Rotifera in winter and spring, except for a large number of Copepoda (Mesocyclops and Undinula) (77.16%). Although the dominant group changed slightly with different markers, it did not affect the trend. In the COI metabarcoding datasets, the zooplankton communities in winter and spring were dominated by Rotifera (67.20%, 59.92%) (Fig. 5F), mainly Brachionus, which contributed 27.92% to the community similarity between the two seasons. In autumn, 78.59% of the sequences were annotated as Copepoda (Fig. 5D), mainly including Eucyclops (contribution rate of 20.44%) (Fig. 7B), but the relative abundances of Rotifera and Cladocera were similar, accounting for 11.49% and 8.95%, respectively. The relative abundance of dominant groups in summer COI showed similar trends to those of 18S rRNA.

Figure 5 The seasonal variation in relative abundance of different zooplankton groups in different metabarcoding datasets.

(A) 18S summer & autumn, (B) 18S autumn & winter, (C) 18S winter & spring; (D) COI summer & autumn, (E) COI autumn & winter, (F) COI winter & spring. Variations from summer to autumn, autumn to winter and winter to spring are shown. Welch’s t-test was used to compare the variation between two adjacent sampling seasons. *, P ≤ 0.05; **, P ≤ 0.01; ***, P ≤ 0.001.

Figure 6 Clustering of the entire zooplankton community assemblages by sampling season using principal coordinate analysis (PCoA) based on the Jaccard dissimilarity index and Bray–Curtis dissimilarity index.

(A) 18S Jaccard dissimilarity index; (B) 18S Bray–Curtis dissimilarity index; (C) COI Jaccard dissimilarity index; (D) COI Bray–Curtis dissimilarity index.

Figure 7 Relative abundance of all samples in different sampling seasons at the genus level.

(A) 18S rRNA; (B) COI.

In general, the entire zooplankton community assemblages showed significant seasonal changes (Fig. 6, PERMANOVA, p < 0.001). In addition, the ANOSIM results further indicated that the zooplankton community had significant seasonal variations (Table S4).

Seasonal changes in zooplankton diversity driven by water environmental factors

The measured values (mean values) of water environmental factors in the Sanmenxia Reservoir during all four seasons (time) in the study period are shown in Table 3. Each water environmental factor showed statistically significant differences between seasons (Kruskal–Wallis, p < 0.01). With the change in sampling season, the WT fluctuated, ranging from 2.0 (winter) to 33.2 °C (summer); the recorded TN concentration showed an increase and reached the highest concentration in spring (5.3 ± 0.16 mg/L); the DO increased from summer and reached the highest concentration in winter (8.31 ± 0.49 mg/L) and then decreased to the lowest concentration in spring (6.16 ± 0.16 mg/L). The pH, COD, TP, and NH4-N showed synchronous changes that fell from summer to autumn, rose again from autumn to winter, and eventually fell from winter to spring.

To understand the environmental factors that drive seasonal changes in zooplankton diversity, we performed partial (geographic distance-corrected) Mantel tests to correlate the DNA metabarcoding datasets and the distance matrix of water environmental factors based on Euclidian distances. The results showed that WT, COD, TN and NH4-N were the main water environmental factors that drive seasonal changes in zooplankton richness (Fig. 8). WT had a significant positive correlation (Spearman’s, p < 0.05) with the richness of most zooplankton groups. Moreover, the water environmental factors that drive seasonal changes in zooplankton groups changed slightly with different groups in the same molecular marker, such as Rotifera and Copepoda. Due to the different taxa and richness identified by different molecular markers, the environmental factors that drive the seasonal changes in these four zooplankton groups were not completely consistent between the two molecular markers.

The CCA results showed that six water environmental factors explained 59.67% and 50.29% of the zooplankton community variance based on the relative abundance in CCA axis 1 and 32.22% and 29.39% in axis 2, respectively (Fig. 9). Among all the factors, WT was the most powerful driving factor, followed by TN (Table S5). However, DO was not found to directly affect the seasonal variation in zooplankton diversity in the current analysis (partial Mantel test, p > 0.05; Adonis, p > 0.05).

Table 3 Water environmental conditions (mean value ± SE) of Sanmenxia Reservoir.

Sample	WT(°C)	pH	COD (mg/L)	TP (mg/L)	NH4-N (mg/L)	DO (mg/L)	TN (mg/L)	
SUM	30.41 ± 0.38	8.11 ± 0.00	48.73 ± 3.71	0.04 ± 0.00	0.45 ± 0.04	6.97 ± 0.67	3.90 ± 0.48	
AUT	15.97 ± 0.42	7.68 ± 0.04	22.98 ± 1.17	0.02 ± 0.00	0.20 ± 0.02	7.77 ± 0.42	4.79 ± 0.03	
WIN	2.67 ± 0.13	8.10 ± 0.05	35.50 ± 3.71	0.07 ± 0.01	0.79 ± 0.05	8.31 ± 0.49	5.05 ± 0.08	
SPR	16.66 ± 1.55	7.81 ± 0.04	30.56 ± 1.69	0.05 ± 0.00	0.23 ± 0.05	6.16 ± 0.16	5.30 ± 0.16	
Notes.

WT water temperature

pH pH

COD chemical oxygen demand

TP total phosphorus

NH4-N ammonium

DO dissolved oxygen

TN total nitrogen

Figure 8 Water environmental factors driving seasonal changes in the zooplankton community.

Pairwise comparison of water environmental factors was achieved by using Spearman tests, and the color gradient represents Spearman’s correlation coefficient. OTUs from different metabarcoding datasets (18S rRNA, COI) were related to water environmental factors by using partial (geographic distance-corrected) Mantel tests. Edge width: Mantel’s R statistic; edge color: statistical significance (based on 9,999 permutations). (A) 18S rRNA; (B) COI.

Figure 9 Canonical correspondence analysis (CCA) of the relative abundance of the zooplankton community in different metabarcoding datasets.

WT, water temperature; pH, pH; COD, chemical oxygen demand; TP, total phosphorus; NH4-N, ammonium; TN, total nitrogen. (A) 18S rRNA; (B) COI.

Discussion

Zooplankton diversity

DNA metabarcoding technology is a powerful tool for large-scale biodiversity research. Multi-gene metabarcoding is a particularly promising approach. A total of 111 families, 174 genera and 190 species were identified in the four sampling seasons by using 18S and COI as markers. Compared with the 48 species (29 species of Protozoa, 16 species of Rotifera, 2 species of Copepoda, 1 species of Cladocera) surveyed in the reservoir area in 1985 (Qiaoyu & Zhifeng, 2005), the number of zooplankton species identified by the metabarcoding method was significantly higher. Based on this significant increase in zooplankton species, we believe that a significant change has occurred in the species composition of zooplankton in the Sanmenxia Reservoir. However, we realize that the previous research used a classification method based on traditional morphology, which was different from the metabarcoding method we used, and the data analysis methods and sampling locations were also inconsistent. Therefore, studies on zooplankton diversity should be conducted by using the same method and sampling location to obtain more comprehensive and reliable results (Li et al., 2019b)

We found that the number of zooplankton species identified by the two molecular markers is larger than that of any single molecular marker in this study. The simultaneous use of two molecular markers can improve the detection of species (Zhang et al., 2018), and make the results of zooplankton diversity research more comprehensive. Different markers not only have different taxonomic resolution but also complement one another (Bucklin et al., 2016; Giebner et al., 2020). The taxa identified by the two markers were different, and we suspect two principal reasons for this difference. Firstly, it could be caused by the different amplification ability of PCR primers for different organisms. Each molecular marker gene has different amplification efficiency for each taxonomic group under the amplification of different universal primers, resulting in different amplified taxa. In this process, PCR amplification bias may be introduced due to universality of the primers, annealing temperature, and the number of replication cycles, etc. (Engelbrektson et al., 2010). Secondly, the use of different reference databases can also lead to differences in taxonomic identification. To more comprehensively identify zooplankton taxa, we need to use both a reference database with taxonomic integrity and that is geographically comprehensive (Bucklin et al., 2016), but we did not choose the NCBI nt database. Instead, we selected the SILVA, and BOLD databases as references for 18S rRNA and COI sequences, respectively. The SILVA database is suitable for annotating 18S rRNA sequences to a more refined taxonomic level compared to the NCBI nt database(Lindeque et al., 2013); the BOLD database is mainly based on COI barcodes and currently contains over 4 million sequences of more than 5 million different species (Wangensteen et al., 2018) Moreover, the purpose of our research was to more comprehensively identify and classify zooplankton diversity rather than compare the diversity and taxonomy of different molecular markers, although the NCBI nt database can facilitate the comparison of multiple molecular markers in the same database (Djurhuus et al., 2018). Therefore, we chose two different reference databases. Interestingly, all 18 classes found only in the 18S rRNA identification results were protozoans, such as Dinophyceae, Phyllopharyngea, and Litostomatea so and on. We think that this sensitivity to protozoans may arise from their closely connection with the PCR primers and reference database we selected. 18S rRNA datasets were mainly used to identify Protozoa in this study, and the PCR primers and reference database of the 18S rRNA datasets we selected were also biased to detect Protozoa, which further indicates that the choice of reference database and primer has a great impact on the detection results. Moreover, it should be pointed out that no matter which database is used, the annotation of the OTU may be biased or blocked due to the absence of the reference sequence(Djurhuus et al., 2018). The 18S marker has a low taxonomic resolution (Gibson et al., 2014), and previous studies have shown that it is not suitable for determining the species-level richness of environmental samples, which was also found in our study (Tang et al., 2012). However, COI is more suitable for taxonomic identification at the species level because it has a higher mutation rate, and its high resolution is usually not achieved by the highly conservative 18S (Dupuis, Roe & Sperling, 2012; Tang et al., 2012). Although the resolution of 18S rRNA genes is lower than that of COI at the species level, its resolution is higher than that of COI for taxonomic levels other than the species level. Therefore, when the research is not aimed at the species level, 18S is sufficient. Otherwise, COI is more suitable. We also found that the lower the taxonomic level, the less overlap between the zooplankton taxa identified by the two markers (Wangensteen et al., 2018). In addition, 18S is superior to COI in identifying Protozoa, but the situation is the opposite when identifying rotifers, copepods and cladocerans. It should be noted that the 18S rRNA datasets only identified 2 OTUs of Cladocera, and neither of them were identified at the species level, which is in sharp contrast with the results of COI identification. Amplicon length polymorphisms have been shown to cause differential bias (amplification and taxonomic) in metabarcoding studies of bacteria and fungi (Ihrmark et al., 2012; Ziesemer et al., 2015), which may explain why only 2 cladoceran OTUs were detected in 18S. If we used only 18S to analyze our samples instead of two markers (nuclear and mitochondrial gene), we would have failed to detect the important influence of cladocerans on the observed beta-diversity (Clarke et al., 2017). This finding emphasized the benefits of using multiple markers metabarcoding methods for biodiversity assessment. It also illustrates that the use of a single molecular marker for the taxonomic identification of environmental DNA is limited. Although inconsistent results may occur by using multiple markers in metabarcoding technology (Dupuis, Roe & Sperling, 2012), it can maximize the information needed for restoring the biodiversity of the ecosystem.

Protozoa, especially ciliates, were the most diverse zooplankton community group, which was consistent with the survey of zooplankton diversity in the Xiangxi River of the Three Gorges Reservoir (Li et al., 2019b). Compared with other zooplankton groups, the protozoan species were dominant, but their relative abundance was low throughout the year, which may be caused by amplification and primer bias. The usefulness of DNA metabarcoding technology for quantitative analysis of species abundance is limited by biological and technical biases, which can affect sequence reads counts (Thomas et al., 2016). The PCR step may introduce chimeric sequences, biasing in taxonomic representation (primer bias) (Bista et al., 2018). Primer-template mismatches can reduce the amplification efficiency, and the resulting bias is further enhanced by the number of PCR cycles (amplification deviation) (Giebner et al., 2020; Piñol, Senar & Symondson, 2019). Moreover, sequencing bias may also affect the abundance estimation in metabarcoding research (Esling, Lejzerowicz & Pawlowski, 2015; Lee et al., 2012). Although the read counts of metabarcoding technology are extremely dependent on the amount of DNA as well as the number of gene copies and primer bias (Elbrecht & Leese, 2015), it can provide a preliminary or semi-quantitative estimation of relative abundance (Yang et al., 2017b). While the relative abundance of reads is not an accurate measure of abundance at the community level, we assumed that these biases would not have a significant impact on intra-taxon comparisons between samples, so these data can be used to examine the comparative differences in the identified taxa between different sampling dates (Banerji et al., 2018).

Rotifers have high values of both richness and relative abundance because they are r-strategists, or opportunists with a small size and short life cycle, and they are tolerant to various environmental factors; thus, rotifers were dominant in the reservoir environment (Neves et al., 2003; Nogueira, 2001). Moreover, the phenotypic plasticity and strong applicability masticatory apparatus of Rotifera also created favorable conditions for its success (Segers, 2007). A large number of studies showed that Rotifera can be used as indicators of water nutrition status, where the presence of Brachionus indicates that water bodies are at moderate to high levels of organic pollution (Sládeček, 1983; Whitman et al., 2004). In our study, Brachionus was found in all zooplankton samples with high diversity, which shows that the water in the reservoir was in a medium or high trophic state. Of course, other genera (Keratella and Rotaria) used to indicate eutrophication status were also found in this study (Mageed, 2007; Tackx et al., 2004). Therefore, it is necessary to strengthen the monitoring of pollution sources in the reservoir area, strictly control the increase of new pollution sources, improve the water quality of the reservoir area, and provide strong support for the sustainable development of water ecology in the reservoir area (Liang et al., 2015)

Seasonal changes in the zooplankton community

Metabarcoding technology can also detect distinct seasonal patterns of taxa that are generally not detectable using traditional methods (Banerji et al., 2018).The analysis shows that the zooplankton community has obvious seasonal changes in Sanmenxia Reservoir. This finding is consistent with previous studies of significant seasonal changes in the zooplankton community based on traditional morphological monitoring and metabarcoding techniques in other freshwater reservoirs (Mwagona, Chengxue & Hongxian, 2018; Nandini, Merino-Ibarra & Sarma, 2008; Tan et al., 2004; Vanderploeg et al., 2012) However, the relative abundance identified for Cladocera did not differ significantly between seasons despite its relative abundance in COI being significantly higher than that in 18S. According to reports, the seasonal changes in Cladocera were controlled by the seasonal changes in nutrients (Xiao-Jun et al., 2014). We found that the seasonal variation in TP was not large in our study, which explains the nonsignificant seasonal variation in Cladocera. Biodiversity patterns were strongly structured temporally, which was reflected in alpha- and beta-diversity measures (Banerji et al., 2018).Their change patterns throughout seasons indicated that zooplankton responded to changes in the water environment of the reservoir. Moreover, when we explored the seasonal variation pattern, we found similar beta-diversity in the two markers, which was consistent with the findings of Clarke et al. (Clarke et al., 2017).

Effects of water environmental factors

Water environmental factors can directly or indirectly affect the zooplankton community. In this study, we found that WT, COD, TN and ammonia nitrogen were the main water environmental factors driving seasonal changes in the richness of the zooplankton community, and six water environmental factors other than dissolved oxygen had important effects on the relative abundance of zooplankton. We also found that WT was the most powerful environmental factor driving the seasonal changes in zooplankton (partial Mantel test, Adonis, p < 0.01) (Sellami et al., 2016; Sunagawa et al., 2015), followed by TN. This result confirmed that temperature and nutrient accumulation were the most important water environmental factors that affected the seasonal changes in the zooplankton community (Li et al., 2019a). Different zooplankton species have specific optimal temperatures (Li et al., 2019a). Appropriate temperature conditions promote the growth and predation of some zooplankton groups, resulting in seasonal changes in zooplankton groups.

Some important variables (chlorophyll a, phytoplankton biomass, particulate organic carbon, precipitation) were not monitored, although we tested multiple water environmental factors. These factors may also have significant effects on the zooplankton community structure, but they exceeded the scope of this study. In addition, it has been observed that climate change and fishery pressure can also cause changes in the populations and composition of zooplankton in aquatic ecosystems around the world (Richardson, 2008; Sarma, Nandini & Gulati, 2005). Unfortunately, understanding these factors is beyond the scope of this study because of the lack of relevant data on climate change, fish farming, and predation in the Sanmenxia Reservoir.

It should be noted that although DNA metabarcoding technology has been widely used in biodiversity research (Deiner et al., 2017; Lim et al., 2016), many technical issues have not been resolved or properly resolved (Xiong, Li & Zhan, 2016a; Xiong et al., 2016b). A large number of studies have extensively discussed the ways in which some technical biases related to DNA extraction, PCR conditions, primer specificity, library preparation and sequence analysis could affect the analysis results (Esling, Lejzerowicz & Pawlowski, 2015; Kermarrec et al., 2013; Lee et al., 2012). Therefore, there is still a great need for a complete and standardized protocol (Majaneva et al., 2018). In addition, the gaps and misidentifications in the reference database can cause obstacles to the assignment of taxonomy to environmental sequences (Visco, 2015). Considering the advantages and limitations of DNA metabarcoding technology, we propose using both 18S and COI as markers to reveal the breadth of zooplankton diversity in the Sanmenxia Reservoir to overcome the inherent limitations of using a single marker.

Conclusion

Zooplankton communities show obvious seasonal variations in the Sanmenxia Reservoir. Temperature and nutrients are the most important water environmental factors affecting the seasonal changes in zooplankton communities. The results of this study provide data support for aquatic biodiversity protection in the Sanmenxia Reservoir and for sustainable ecological management.

Supplemental Information

Supplemental Information 1 Amplification primers for 18S rRNA and COI

Click here for additional data file.

Supplemental Information 2 The sequencing depth of the two molecular markers

Click here for additional data file.

Supplemental Information 3 The Non-plankton in the metazoan identified by the DNA metabarcoding

Click here for additional data file.

Supplemental Information 4 Analysis of similarities (ANOSIM) showed seasonal variation in zooplankton community based on Jaccard and Bray–curtis distance

Click here for additional data file.

Supplemental Information 5 Adonis analysis shows the contribution and significance of each water environmental factor to the seasonal changes in the relative abundance of zooplankton. *, p < 0.05; **, p < 0.01; ***, p < 0.001

Click here for additional data file.

Supplemental Information 6 Sampling sites for zooplankton and water samples in Sanmenxia Reservoir

Click here for additional data file.

Supplemental Information 7 Rarefaction curve for each sample based on 18S rRNA (A) and COI gene (B)

Click here for additional data file.

Supplemental Information 8 The seasonal distribution of zooplankton species is sample-based. (A) 18S rRNA; (B) COI

Click here for additional data file.

We thank Yingchun Lu for assisting us in sample collection, Hao Yuan, Xiaopeng Wang, Yuanquan Luo and Houyun Huang for our help in data analysis, and Xiaolei Feng for improving the language of the manuscript.

Additional Information and Declarations

Competing Interests

Author Contributions

Field Study Permissions

DNA Deposition

Data Availability

The authors declare there are no competing interests.

Lina Zhao and Ying Mao conceived and designed the experiments, performed the experiments, analyzed the data, prepared figures and/or tables, authored or reviewed drafts of the paper, and approved the final draft.

Xue Zhang conceived and designed the experiments, performed the experiments, analyzed the data, prepared figures and/or tables, and approved the final draft.

Mengyue Xu analyzed the data, authored or reviewed drafts of the paper, and approved the final draft.

Yuan Huang conceived and designed the experiments, authored or reviewed drafts of the paper, and approved the final draft.

The following information was supplied relating to field study approvals (i.e., approving body and any reference numbers):

Field experiments have obtained verbal permission from the Sanmenxia Reservoir Management Office. Verbal permission from the Sanmenxia Reservoir Management Office was provided by the reservoir area security guards (anonymous, name unknown). However, because our sampling location belongs to public open access, no permission is ultimately required.

The following information was supplied regarding the deposition of DNA sequences:

Data are available at NCBI SRA: SRR12007207–SRR12007278.

Data is also available at Figshare:

Zhao, Lina; Zhang, Xue; Mao, Ying; Huang, Yuan (2020): Species diversity and seasonal variation in zooplankton in the Sanmenxia Reservoir characterized by the 18S rRNA and COI metabarcoding methods. figshare. Dataset. https://doi.org/10.6084/m9.figshare.12495914.v2.

The following information was supplied regarding data availability:

Raw data and water environmental factors for this article are available at Figshare:

Zhao, Lina; Zhang, Xue; Mao, Ying; Huang, Yuan (2020): Species diversity and seasonal variation in zooplankton in the Sanmenxia Reservoir characterized by the 18S rRNA and COI metabarcoding methods. figshare. Dataset. https://doi.org/10.6084/m9.figshare.12495914.v2.

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
