# Peer review of "DNA metabarcoding of zooplankton communities: species diversity and seasonal variation revealed by 18S rRNA and COI"

_PeerJ, doi:10.7717/peerj.11057_

## Round 0.1 · original submission · Major Revisions

All three reviewers think that the data presented in the manuscript are interesting and will deserve publication. However, we also agree that the manuscript would greatly benefit from a series of major improvements.

Please read carefully the following reviews and modify your manuscript accordingly. I will be looking forward to an exhaustive review, addressing all major issues pointed out by the reviewers. I think that the following particular points are crucial:

1. The manuscript would benefit from a change in the title and a re-framing of abstract and introduction, so that it can be attractive to a broader audience, and not just to "those interested in the Sanmenxia Reservoir zooplankton".

2. The Methods section is especially problematic. All reviewers agreed that the bioinformatics procedures are not fully explained, the original data are not publicly available and the methods are hardly reproducible. The authors have to explain details of the quality filtering steps and, specifically, provide more details about the taxonomic assignment procedures. It could be a good idea to use a more advanced algorithm for taxonomic assignment, such as the ones suggested by the reviewers, and not to rely exclusively in the results of a BLAST search.

3. The manuscript would greatly benefit from a deeper discussion about differences in the results from using two different metabarcoding markers. All reviewers agree that it would be very interesting to include a more explicit comparison of the detectability of different taxonomic groups by using both markers. Moreover, it is not clear how the results from both markers were integrated. Were 18S and COI MOTUs simply summed up together to build a single dataset? Please explain details of how you dealt with duplicity of detected MOTUs for different taxonomic groups.

4. Also, all reviewers suggested that the manuscript should include a more exhaustive review of previously published articles on metabarcoding of freshwater zooplankton, in order to compare the newly presented results with those from previous works.

5. Please, feel free to expand the Supplementary Materials, so you can add more valuable and detailed information about informatics procedures and results of supplementary analyses and comparisons while keeping the manuscript length within reasonable limits.

Reviewer 1 ·

Basic reporting

The manuscript would be improved by careful editing for English throughout.

Experimental design

The main purposes of the paper as stated at the end of the Introduction are to explore the composition of the Sanmenxia zooplankton, the seasonal dynamics and the environmental factors driving the seasonal changes. Based on this, the only people who will want to read this paper are those interested in the Sanmenxia Reservoir zooplankton, a very narrow audience! The paper could be re-framed to provide advice on how future zooplankton metabarcoding studies should be carried out. For example, the abstract mentions this study highlighted limitations and advantages of using two metabarcoding markers for zooplankton diversity. Currently this is not covered in the Discussion, but would be a useful addition to the paper. For example, was the 18S marker better at detecting protozoa, or did COI do a good job too? Would you recommend using both markers for future studies, given the two markers detected essentially the same patterns? If not, which marker would you choose? This could be based on the level of taxonomic detail each marker provided (but see comment on taxonomic accuracy below).
Alternatively, are there broader management implications for these results? The authors state several taxa indicative of eutrophication are present (L376-379), should efforts be made to reduce organic pollutants in the Sanmenxia Reservoir? Large sections of the Discussion currently repeat (or belongs in) the Results (e.g., L386-412), the Discussion could be better used to place the results in context and discuss the implications for future metabarcoding studies.
The taxonomic assignment of OTU sequences needs to be repeated with a more rigorous approach. Was any effort made to confirm the taxonomy was accurate? A ‘top-hit’ approach is prone to errors. Programs such as MEGAN are useful to ensure taxonomic assignments reflect uncertainty, particularly important for markers like 18S with good taxonomic breadth but poor taxonomic resolution. This is particularly relevant for the comparison with historical data based on morphology (L342-345).

The methods section does not provide enough detail to replicate the results. For example, how long was the plankton net dragged for? What volume of water was sampled for the laboratory analyses (chemical oxygen demand etc.)? What MiSeq kit was used, and with how many base pairs? How was the bioinformatic quality control done, e.g., what were the settings used? The first data processing step is converting the FASTQ files to FASTA, suggesting it was not based on the sequencing quality scores. Line 162 suggests replicate samples were collected, this should be detailed in the section on zooplankton collection.

Validity of the findings

No comment

Additional comments

The authors present a study on seasonal changes in zooplankton diversity in the Sanmenxia Reservoir based on metabarcoding data from 18S rRNA and mitochondrial COI. Winter and spring zooplankton communities were similar, but summer and autumn communities were distinct. These changes were related to water temperature and other environmental variables, with both markers showing similar trends. The number of samples collected across many parts of the reservoir suggest the results observed are robust.
Much of the paper assumes that differences in read abundance for each marker reflect the abundance of the zooplankton OTUs. This assumption may not be accurate (in contrast to the statement on reliably detecting relative abundance on L103) and needs to be acknowledged. The authors should also clarify whether they used the binary Jaccard distance (as opposed to ‘Jaccard’.
The reason some of the statistical tests were chosen is unclear. For example, a parametric test (ANOVA) is used for the zooplankton group richness, but a non-parametric test for overall zooplankton diversity (Kruskal-Wallis). Welch’s t-test is used to look at changes over the four seasons, but is designed for testing the difference between two samples (in this case seasons). A different test would be more appropriate.
The manuscript would be improved by careful editing for English throughout.
Minor comments:
L69-70: This sentence is not connected to the rest of the paragraph. Rather, it sounds like you need a new method to look at zooplankton diversity.
L114: Many studies have used more than one marker to look at zooplankton diversity (whether fresh or marine), including Berry et al. (2019, cited above) and the ‘Tree of Life’ approach developed by Stat et al. (2017).
L116: I suggest the main advantage of using more than one marker is the increase in taxonomic breadth, which is especially important for diverse groups like zooplankton. For example, the COI marker used in this study is designed for metazoans but not protozoa, whereas the 18S rRNA marker should detect most protozoan groups.
L128: No hypothesis is stated.
L153: Why were samples ‘filtered through’ 0.22 um filter paper? Was this to remove water?
L216-217: It would be useful to provide the range of sequencing depths separately for both markers.
L220: The singletons should be removed prior to creating rarefaction curves.
L226 and 234: Only 714 of ~1600 OTUs were zooplankton, and only a small percentage were unclassified, what were the remaining >800 OTUs? Similar for the 10-15% of sequences that were not classified as 18S or COI.
L263 and throughout: The division of zooplankton groups used is odd, as Protozoa are not a clade, rotifers are a phylum and cladocera and copepod are classes. However, if you state this is the division you are using in the methods it can be accepted.
L295: If this is the main figure for this result, it should be in the main text.
L344: References should be provided for at least the 1985 study.
L376: Suggest changing Brachiosaurus to Brachionus.
L427-428: Relative abundance always sums to 100%, so if everything else goes down, rotifers would have to increase.
L452: This statement needs references.
L457: Do you mean OTU-level, given that is what is shown in Figure 8?
Figure 1: Change to bar charts.
Figure 3: Define A and B in legend.
Figure 5: These could be combined into two panels instead of 6. Does the x-axis shopw the mean relative abundance per sample? If so, the standard deviation or standard error should also be shown.
Figure 6: Are the ellipses 75% confidence intervals? This should be included in the legend.
Figure 8: I’m not sure panel A is informative. It is not referred to in the text.
Figure 9 is not referred to in the Results (but is in the Discussion). I suggest reducing the number of taxa shown to 10-15 at most, as it’s difficult to discern the differences between many shades of the same colour. Also, taxa could be colour-coded by zooplankton group, e.g. rotifer genera are all shades of grey, copepods are shades of blue etc.

Reviewer 2 ·

Basic reporting

Article structure can be improved. For example, not all figures are explained in the Result Section. Labels in some figures are too small and not clear, and readers cannot understand results authors present. Raw data of sequence reads are available; however, each data is not fully explained. In the current version of the data deposited, it is very difficult to be used secondary in the future. I also recommend to use database in NCBI BioProject, in order to promote secondary uses of authors’ data. I’m not native speaker, and I cannot evaluate English in the manuscript.

Experimental design

There are significant flaws in the method section.

Methods are poorly explained, and there is insufficient information to be reproducible by another investigator. In particular, detailed information of bioinformatic analysis are not provided, and I couldn’t fully evaluate accuracy of results in the manuscript. Authors should explain details of quality filtering steps. Also, they should clarify how they annotated OTUs into taxonomic groups (species, genus or other taxonomic level). There are also no information if authors have DNA barcoding data for species in the study area.

Primers for COI gene are mainly constructed for Metazoa. In my impression, these primers couldn’t evaluate protozoa accurately. In addition, 18S is relatively conserved for species-level. Authors present data with combination of two markers (18S + COI). However, because of these biases, these markers should be analyzed separately. I also doubt that OTUs derived from a same species present in analyses of 18S + COI. Biases between makers are clearly visualized in Figure 1.

I found OTUs classified into marine taxa. I think taxonomic classification is not properly performed in the manuscript, or there are insufficient database for 18S or COI. Anyway, I couldn’t judge cause of mis-identification, because of lacks of information in Methods.

Authors removed a singleton read of OTUs. So diversity cannot be evaluated by Chao 1 index.

Validity of the findings

Although results are descriptive, data obtained in this study is valuable. The metabardcoding data can be compared in the future under different environmental conditions. However, as described above, accuracy of data couldn’t be fully evaluated because of insufficient explanations in the method section.

Reviewer 3 ·

Basic reporting

- English could be improved in some parts of the manuscript.

- General lack of references that would allow comparing the results to previous studies on freshwater zooplankton

Experimental design

- Sampling, laboratory workflow and bioinformatic processing should be described in more detail. Replication would not be possible with the information provided at the moment.

Validity of the findings

- Results should e compared to previous findings. It is not clear if the work is highly novel or if similar studied have been conducted on other reservoirs.

Additional comments

The authors conducted an interesting study on freshwater zooplankton. I am sure that the results merit publication, but several aspects of the manuscript suffer from a lack of references, and the methods used should be described in more detail to allow other researchers to reproduce results/adopt protocols. I therefore recommend major revisions.

Abstract:
18 - 21: You might want to split this sentence in two.

22: Please mention that COI is a mitochondrial gene

31: It is not clear what 'biological interaction' means in this context.
Please specify

Introduction:

93: There are several publications on bias introduced by morphological
identification of invertebrates, e.g. Haase et al. 2006. Please cite at least one or two.

103: The statement that metabarcoding can reliably detect abundances is debatable and not necessarily true for all environments and all taxonomic groups, and also depends on the genetic marker and library preparation method used. Please tone down this statement or clarify which taxonomic groups and ecosystems you mean.

113: Please cite papers which show the limitations of single marker approaches regarding the detection of certain taxonomic groups

119-125: You explain what you expect, but your expectations are probably based on previous observations or patterns known from other freshwater ecosystems. Please provide some references.

132: Please provide a reference for metabarcoding using the 18S v4 region, as readers without a background in molecular biologiy do not necessarily know what the v4 region is.

133: Please mention the full name of COI and that it is a mitochondrial gene

General remark: Is there any data on the taxonomic diversity of the Sanmenxia zooplankton or is your study the first to look into this? Is there any data on zooplankton communities from similar dams in the area? If so, please cite previous work. If not, please explicitly mention this.

Material and methods

141: When was the dam completed?

145: Which geographical features were used to decide on sampling locations?

150: Was the sampling standardised by sampling identical volumes? If not, did oyu correct for this?

153: Why did you filter the samples?

158: Did you collect additional water sampled for these measurements?

162: Why did you decide to pool different samples?

168: Please elaborate on the workflow and mention all library preparation steps or cite the protocol that you followed.

173: Did you use the PEAR default settings?

174: Trimmomatic: Which settings were used?

177: Which 'additional quality filtering' steps were conducted using USEARCH?

182/183: How did you proceed if an OTU could not be assigned to a species name? Did you assign the OTU on higher taxonomic level or did you exclude it from downstream analyses?

General remarks:
Did you normalise your data by subsampling?
Did you remove samples with extremely low read numbers (if there were any)?
Did you process biological or technical replicates?
Did you run negative controls to check for contamination?
Did you take into account that OTU clustering both 18S and COI with 97% similarity could lead to a bias,as these markers do not have the same evolutionary history and mutation rates?

Results

Please provide some basic information on the number of obtained reads and if any samples had to be excluded because of low read count or low quality.

If I understood correctly, you pooled OTUs obtained through 18S and COI metabarcoding. However, this means that some species were present with two or more OTUs (one in 18S, one in COI). Some species might have een amplified using the 18S primers, but not the COI primers or vice versa. Did you somehow account for that? If not, this could greatly bias your data. If you did account for that or used a different approach, please explain this in more detail to make it clear.

252: Please specify which group was the most abundant. Also keep in mind that this result could be a result of primer and/or amplification bias. Please discuss this carefully. Maybe you can cite literature showing that that copepods and/or rotifers are commonly the most abundant taxa in reservoirs?

320-330: Did you apply a Bonferroni correction to account for multiple testing?

329: Please elaborate on the different results you found for the two markers

Discussion

344: Please provide the references for the surveys conducted in 1985 and 2005

350-351: Do you mean 'Therefore, studies on zooplankton diversity can only be conducted by using the same method and sampling location, which allows obtaining comprehensive and reliable data and comparing results'?

356: Please provide a references for the 1986 study

359: Do you mean'the Protozoa were dominant'?

360: 'disadvantage' is probably not the right word here. Please rephrase.

361: Can you be sure that predation was the main factor driving protozoan abundance? It could also be a change in biomass ratio caused by other factors, which can lead to different sequencing results. Please discuss potential amplification and primer bias.

362: I assume you want to say that they are filter feeders? 'fed filters' is not correct.

367: Are there studies showing how incomplete the reference databases for Protozoa are?

371: Isn't a reservoir a relatively stable, non-dynamic environment?

377: Diversity in terms of OTU number?

381-382: This sentence should be rephrased

385: Other freshwater reservoirs, or what kind of location?

387-388. Did you expect this? If so, why? Also, please discuss why you found differences between 18S and COI datasets.

397-412: Many of these numbers could go to the Results. Also, please discuss them and compare your findings to previous studies. More references would help understand what your results actually mean.

415-417: Were any of these factors autocorrelated? Can you say that all were relevant, or could it be that only one or few factors were the actual drivers?

422: Please provide references for the statement that different groups have different temperature optima.

450-452: Many statements without references.

462-467: Please rephrase and split into two or three sentences.

---

## Round 0.2 · Minor Revisions

As pointed out by the reviewer, the present version of the manuscript has been considerably improved, and its scientific quality makes it now suitable for being published in PeerJ.
However, I agree with the reviewer that the English language used in the manuscript is still somehow awkward in many parts of the paper. An improved version, possibly corrected by someone with a good level of written English, should be produced and resubmitted. I consider this request for correcting the English as "minor changes". I am not providing a detailed list of sentences and expressions to improve here, since they are way too many and they are distributed all along the manuscript, including figure captions. Basically, the manuscript needs a full review and re-editing of the English text, in order to reach the minimum quality standards of readability.
As an example, I have re-written some parts of the abstract. Here is my suggestion for a corrected version of the abstract:
"Background. Zooplankton is an important component of aquatic communities and has important biological and economical significance in freshwater ecosystems. However, traditional methods that rely on morphology to classify zooplankton require expert taxonomic skills. Moreover, traditional classification methods are time-consuming and labour-intensive, which is not practical for the design of conservation measures and ecological management tools based on zooplankton diversity assessment.
Methods. We used DNA metabarcoding technology with two different markers: the nuclear small subunit ribosomal RNA (18S rRNA) and mitochondrial cytochrome c oxidase (COI), to analyze 72 zooplankton samples collected in 4 seasons and 9 locations from the Sanmenxia Reservoir. We investigated seasonal changes in the zooplankton community and their relationship with water environmental factors.
Results. A total of 190 species of zooplankton were found, belonging to 12 phyla, 24 classes, 61 orders, 111 families, and 174 genera. Protozoa, especially ciliates, were the most diverse taxa. Richness and relative abundance of zooplankton showed significant seasonal changes. Both alpha and beta diversity showed seasonal trends: the diversity in summer and autumn was higher than that in winter and spring. The zooplankton diversity was most similar in winter and spring. By correlating metabarcoding data and water environmental factors, we proved that water temperature, chemical oxygen demand, total nitrogen and ammoniacal nitrogen were the main environmental factors driving the seasonal changes in zooplankton in the Sanmenxia Reservoir. Water temperature, followed by total nitrogen, were the most influential factors. This study highlights the advantages and some limitations of zooplankton molecular biodiversity assessment using two molecular markers."

Also, please be extremely careful with all names of zoological taxa (including those written in figures and tables). For example, I have detected the use of "Euclops macruroides" instead of "Eucyclops macruroides", and the use of "Rorifera" instead of "Rotifera" in the caption of Fig. 1. Please do a final check of them all to look for more typos.

I am looking forward very much to receive such a corrected version of your highly interesting and relevant paper, with improved readability, and I can foresee that it will be promptly accepted for its publication in PeerJ.
Thank you.
Owen S. Wangensteen

Reviewer 3 ·

Basic reporting

While the text has improved, the authors might want to ask a native English speaker to proofread the manuscript.

Experimental design

No comment

Validity of the findings

No comment

Additional comments

The authors addressed all concerns raised during the first round of review and rewrote large parts of the manuscript. The results are now presented more carefully, and limitations of the dataset are described and discussed. I feel that the manuscript and the reported datasets are interesting and should be published. However, the authors might want to ask a native English speaker to proofread their manuscript before publication, as some parts still read rather awkwardly.

---

## Round 0.3 · Minor Revisions

Dear Dr. Zhao,
I think that the language in the manuscript has been greatly improved. However, I still think that some sentences must be corrected, for a better readability. Please find the following list of suggested changes. I am sure that it will take you very little time to make these modifications. When you resubmit the manuscript with these, then I will promptly accept it for publication in PeerJ.

L46: Correct "to" in "are limited to North America". Also, this sentence needs a citation; for example: Ricciardi A & Rasmussen JB. Extinction rates of North American freshwater fauna. Conservation biology. 1999 Oct 23;13(5):1220-2.
L48: "However, protection requires a detailed understanding of this biodiversity."
L108: "can increase the taxonomic breadth"
L109: "the biodiversity of highly diverse groups such as zooplankton."
L113: "we hypothesize that the Sanmenxia Reservoir"
L119: "To verify the above hypothesis"
L122: "We used a multigene approach with two independently evolved markers,the nuclear small subunit ribosomal RNA (18S rRNA v4 region; Hadziavdic et al. 2014) and the mitochondrial cytochrome c oxidase (COI; Leray & Knowlton 2015)."
L138: "Zhang"
L146: "at a depth of 0.3 m".
L161: "was purified by"
L162: remove the d before the parenthesis in "(Qiagen, Germany)"
L183: "a 50 bp window size was set, and bases with average Phred quality score less than 20 were trimmed."
L185: Which software did you use to remove the primer sequences? It should be mentioned here, either before of after paired-end merging, depending on the particular pipeline you used.
L232: Correct the number! "2,185,317 and 910,137"
L232: "high-quality reads (including non-plankton metazoans) were obtained from the 18S rRNA and..."
L281-282: Check the rephrasing: "Copepoda had significantly higher relative abundance than other zooplankton taxa for both genetic markers, and Cladocera had the lowest relative abundance."
L319: "Rotifera"
L413: "The 18S marker has a low taxonomic resolution (Gibson et al. 2014),..."
L419: "its resolution is higher"
L431: "on the observed beta-diversity"
L453: "is not an accurate measure of abundance at the community level,"
L457: "Rotifers have high values of both richness and relative abundance"

Looking forward to have your corrected final version soon.
Congratulations,
Owen Wangensteen

---

## Round 0.4 · accepted · Accept

Congratulations for this interesting and relevant work.